# A New Beam Model for Simulation of the Mechanical Behaviour of Variable Thickness Functionally Graded Material Beams Based on Modified First Order Shear Deformation Theory

**DOI:** 10.3390/ma12030404

**Published:** 2019-01-28

**Authors:** Vu Hoai Nam, Pham Van Vinh, Nguyen Van Chinh, Do Van Thom, Tran Thi Hong

**Affiliations:** 1Division of Computational Mathematics and Engineering, Institute for Computational Science, Ton Duc Thang University, Ho Chi Minh City 700000, Vietnam; vuhoainam@tdtu.edu.vn; 2Faculty of Civil Engineering, Ton Duc Thang University, Ho Chi Minh City 700000, Vietnam; 3Department of Mechanics, Le Quy Don Technical University, Hanoi City 100000, Vietnam; phamvanvinh.mta@gmail.com (P.V.V.); ngchinhhd@gmail.com (N.V.C.); 4Center of Excellence for Automation and Precision Mechanical Engineering, Nguyen Tat Thanh University, Ho Chi Minh City 700000, Vietnam

**Keywords:** Model, modified first order shear deformation, variable thickness beam, FGM, finite element method

## Abstract

There are many beam models to simulate the variable thickness functionally graded material (FGM) beam, each model has advantages and disadvantages in computer aided engineering of the mechanical behavior of this beam. In this work, a new model of beam is presented to study the mechanical static bending, free vibration, and buckling behavior of the variable thickness functionally graded material beams. The formulations are based on modified first order shear deformation theory and interpolating polynomials. This new beam model is free of shear-locking for both thick and thin beams, is easy to apply in computation, and has efficiency in simulating the variable thickness beams. The effects of some parameters, such as the power-law material index, degree of non-uniformity index, and the length-to-height ratio, on the mechanical behavior of the variable thickness FGM beam are considered.

## 1. Introduction

Beams are used widely in engineering fields, such as aerospace, mechanical, and civil engineering, and so on. Researchers have proposed various analytical and numerical methods to analyze static bending, free vibration, and buckling of beams. Chakrabortyet al. [1] developed a new beam element model based on first order shear deformation theory (FSDT) for static bending, free vibration of thermo-elastic FGM beams. Khan et al. [2] simulated the static bending and free vibration analysis of FGM beams using the finite element method (FEM) and zig-zag theory. Wang et al. [3] and Lee et al. [4] applied Euler-Bernoulli beam theory to analyze the free vibration of FGM beams. The nonlinear bending of a two-directionally FGM beam was considered by Li et al. [5] using the generalized differential quadrature method (GDQM). Jafari et al. [6] used analytical approximation for nonlinear vibration analysis of Euler-Bernoulli beams. Alshorbagy et al. [7] used FEM to analyze free vibration of a Euler-Bernoulli beam made of FGM. Oreh et al. [8] employed FEM for stability and free vibration analysis of a Timoshenko beam. Li et al. [9] applied an analytical solution to investigate relations between buckling loads of functionally graded Timoshenko and homogeneous Euler-Bernoulli beams. Farhatnia et al. [10] used first order shear deformation theory and GDQM for buckling analysis of FGM thick beams. Kien et al. [11] used the analytical solution for static bending, free vibration, and buckling analysis of axially loaded FGM beams. Sina et al. [12] used FSDT and the analytical solution for free vibration analysis of FGM beams. To avoid shear correction factor in the first order shear deformation, the higher-order shear deformation theory (HSDT) was developed. Static bending of FGM beams was investigated by Kadoli et al. [13] using HSDT and FEM. Thai et al. [14] applied the analytical solution and HSDT for bending and free vibration analysis of FGM beams. The nonlinear bending of FGM beams was studied by Gangnian et al. [15] using HSDT and the differential quadrature method. Celebi et al. [16] applied different beam theories for free vibration analysis of FGM beams. Hadji et al. [17] used exponential shear deformation theory and the analytical solution to study static bending and free vibration of FGM beams. Based on different higher-order beam theories, Simsek [18] analyzed the fundamental frequency of FGM beams. Ghumare et al. [19] developed a new fifth-order shear and normal deformation theory to analyze static bending and elastic buckling of P-FGM beams. Li et al. [20] used higher-order theory for static and dynamic analyses of functionally graded beams. Li et al. [21] have developed an exact frequency equation for free vibration analysis of axially FGM beams. According to the mentioned studies, although HSDT has overcome the shear locking phenomenon without the correction factor, it increases the number of degrees of freedom, has complex formulations, and still needs reduced integration. In addition, due to the simplicity of FSDT, it has been modified and improved by many scientists, so that a modified first order shear deformation is employed in this study to developed a new beam model.

In practical projects, when using beams with variable thickness, the system not only is lighter, but also the aesthetics of the structures can be enhanced. Motaghian et al. [22] developed a new Fourier series solution for free vibration analysis of non-uniform beams resting on elastic foundation using Euler-Bernoulli beam theory. Shahba et al. [23] applied the differential transform element method and differential quadrature element method of lowest order to simulate the free vibration analysis of tapered Euler-Bernoulli beams made of axially FGM. The large displacement of a tapered cantilever FGM beam was studied by Kien [24,25] using FEM. Hassanabadi et al. [26] investigated free and forced vibration of non-uniform isotropic beams using orthonormal polynomial series expansion. Tong et al. [27] developed an analytical solution for free and forced vibrations analysis of stepped Timoshenko beams and used approximate analysis of generally non-uniform Timoshenko beams. Tang et al. [28] developed an exact frequency equation for free vibration analysis of exponentially non-uniform functionally graded Timoshenko beams. Huang et al. [29] analyzed the free vibration of axially FGM Timoshenko beams with a non-uniform cross-section using a power series function. Calim [30] applied the complementary functions method for transient analysis of axially FGM beams. Abadi et al. [31] investigated the free vibration of variable cross-section beams using the asymptotic solution. Xu et al. [32,33] provided elasticity solutions to analyze the static bending of a variable thickness beam and multi-span beams with variable thickness under static loads. Banerjee et al. [34] applied the dynamic stiffness method to study free vibration of rotating tapered beams. Zenkour [35] studied the elastic behavior of a variable thickness beam using the analytical solution and FSDT. Lin et al. [36] analyzed geometrically nonlinear bending of variable thickness FGM beams using the meshless method. Nevertheless, the analytical method is an inefficient and complex method for studying variable thickness structures, while FEM proves a convenient, simple, and highly effective method. However, locking can occur with the basic element types when using FEM. To overcome shear locking, reduced integration was used, but reduced integration can cause unwanted behavior, which is the spurious mode or hour-glassing in a mesh of multiple elements [37]. Another way to prevent shear locking is the method of incompatible modes. The incompatible mode elements can be formulated as low-order enhanced strain elements, in which strains (those corresponding to the incompatible displacements) are added to the usual strains derived from the compatible displacements. Although the incompatible mode elements are best suited to model pure bending, incompatible mode elements can cause spurious modes when geometrically nonlinear displacement analysis with small displacements and small strain conditions [38]. On the other hand, the use of incompatible mode elements can provide difficulties in large strain analysis; for such an analysis, the displacement/pressure (or u/p formulation) elements are more reliable and effective [37,39].

FGM is microscopically inhomogeneous composites fabricated from a mixture of ceramics and metals. FGM plates, beams, and shells can be considered as a multi-coated thin-walled composite mechanical components structures, making it possible to obtain specific mechanical properties, such as superior stiffness-to-weight ratio, high strength, and high damping. The damping efficiency of the coating systems was discussed by some researchers [40,41,42,43,44,45]. In these works, the theoretical and experimental results show that the coating structure obtained the best balance between the strength and the damping capacity, which is important for reducing the mechanical vibrations. Due to its outstanding features, the FGM has been extensively used in engineering applications, including nuclear power plants, spacecraft, turbine engines, and automotive and biomechanical applications [46,47,48,49,50,51]. Because of an increasing use of the FGM for many engineering applications nowadays, further studies for mechanical behaviours of structures made of FGM are necessary.

The main purpose of this contribution is to develop a new beam model based on modified first order shear deformation theory. The proposed beam model is simple in its formulation and free of shear locking without reduced integration. The proposed beam model is used for static bending, free vibration, and buckling analysis of a variable thickness beam made of FGM to demonstrate its accuracy and effectiveness.

The organization of this paper is as follows. In Section 2, a brief review of first order shear deformation theory is presented. In Section 3, a modified first order shear deformation for the beam model is given. Next, in Section 4, based on modified first order shear deformation and interpolating polynomials, a new beam model is developed. In addition, the element stiffness matrix, element mass matrix, and element force vector are constructed. Section 5 focuses on verification and numerical analysis for static bending, free vibration, and buckling responses of variable thickness FGM beams. In addition, the effects of some parameters on the mechanical behavior of the FGM beam are investigated. In the conclusion section, some discussions and highlights of the proposed beam model are given.

## 2. Basic Equations of First Order Shear Deformation Theory of a Timoshenko Beam

The axial displacement, u, and the transverse displacement, w, at any point based on FSDT of a Timoshenko beam are given by:(1)$$\left\{ \begin{array}{l} u(x,y,z)=-z\psi (x)\\ w(x,y)=w_{0}(x) \end{array}\right.$$ where $w_{0}$ is the transverse displacement at the middle axial beam, and ψ is the angle of cross-section rotation.

In small deformation, we have:(2)$$\frac{\partial w}{\partial x}=\psi +\theta$$

The bending moment resultant and shear force are taken by:(3)$$M=D\frac{\partial \psi }{\partial x},\;Q=S\left(\frac{\partial w}{\partial x}+\psi \right)$$ in which:(4)D=EI, S=kGA is flexural rigidity and shear rigidity, respectively, E is the Young’s modulus and G=E/[2(1+ν)] is the shear modulus, ν is Poison’s ratio, k is the shear correction factor, and *I* and *A* are, respectively, the second moment of area and cross-section area of the beam.

Equilibrium of moment and transverse forces leads to:(5)$$\frac{\partial M}{\partial x}-Q=0,\;\frac{\partial Q}{\partial x}=0$$

Substituting Equations (3) and (4) into (5) results in:(6)$$\begin{array}{l} \frac{D}{S}\frac{\partial^{2}\psi }{\partial x^{2}}-\left(\frac{\partial w}{\partial x}+\psi \right)=0,\\ \frac{\partial^{2}w}{\partial x^{2}}+\frac{\partial \psi }{\partial x}=0 \end{array}$$

## 3. Modification First Order Shear Deformation for the New Beam Model

Assuming that the total deflection consists of bending deflection and a contribution of transverse shear, the angles of the cross-section slope are the result of pure bending angles and shear angles as shown in Figure 1:(7)$$w=w_{b}+w_{s},\;\psi =-\frac{\partial w_{b}}{\partial x}+\theta$$ the subscripts, *b* and *s,* denote the bending and shear deflection, respectively.

By substituting Equation (7) into Equation (6) it is possible to separate the variables of two different displacement fields:(8)$$\frac{\partial }{\partial x}\left[\frac{D}{S}\frac{\partial^{2}w_{b}}{\partial x^{2}}+w_{s}\right]=\frac{D}{S}\frac{\partial^{2}\theta }{\partial x^{2}}-\theta$$

(9)$$\frac{\partial^{2}w_{s}}{\partial x^{2}}=-\frac{\partial \theta }{\partial x}$$

Equation (8) and Equation (9) have three unknown components, $w_{b},\;w_{s}$, and θ, are satisfied if ${w^{\prime }}_{b}=-\theta$ and ${w^{\prime }}_{s}=\theta$. That implies $w_{b}=-w_{s}$ and $w=w_{b}+w_{s}=0$, that solution is trivial since θ is an arbitrary function. Therefore, Equation (8) and (9) have a realistic solution when:(10)$$\frac{D}{S}\frac{\partial^{2}w_{b}}{\partial x^{2}}+w_{s}=0$$

Then:(11)$$w_{s}=-\frac{D}{S}\frac{\partial^{2}w_{b}}{\partial x^{2}}$$

Substituting Equation (11) into Equation (9), the total deflection is:(12)$$w=w_{b}-\frac{D}{S}\frac{\partial^{2}w_{b}}{\partial x^{2}}$$

## 4. The Variable Thickness FGM Beam Element Model

The variable thickness FGM beam with length, L, as depicted in Figure 2 is considered.

The height of the beam varies along the axial of beam: (13)h=h(x)

The material of the beam is made of two partial materials, those are metal and ceramic. The ratio of values of materials assumes that it varies through the z-direction with the power-law [46,47,48,49,50]:(14)$$V_{c}={\left(\frac{z}{h}+\frac{1}{2}\right)}^{p},\;V_{c}+V_{m}=1$$ where $V_{c},\;V_{m}$ are, respectively, the volume fraction of the ceramic and metal, p is the gradient index of the volume fraction, and h=h(x) is the thickness of the beam. The subscripts, c, m, denote the ceramic and metal constituents, respectively.

The effective material properties, P, such as Young’s modulus, E, and mass density, ρ are expressed using the rule of mixture [46,47,48,49,50]:(15)$$P(z)=\left(P_{c}-P_{m}\right)V_{c}+P_{m}$$

The Poisson’s ratio, ν, is assumed constant in this study.

A two-node beam element with two degrees of freedom per node is expressed here. The bending deflection may be expressed as follows:(16)$$w_{b}=P_{b}a$$ where a is the coefficient vector with unknown terms $a_{i}$ of the approximation polynomial, and $P_{b}$ is bending polynomial, which may be demonstrated as:(17)$$P_{b}=\left[\begin{array}{cccc} 1 & \xi & \xi^{2} & \xi^{3} \end{array}\right]$$ where $\xi =\frac{2x-l_{e}}{l_{e}}$ is a non-dimensional coordinate. 

The shear deflection is taken by:(18)$$w_{s}=P_{s}a$$

According to (11), the terms of shear polynomial are:(19)$$P_{s}=-\left[\begin{array}{cccc} 0 & 0 & 2\alpha & 6\alpha \xi \end{array}\right]$$
(20)$$\alpha =\frac{4D}{Sl_{e}^{2}}$$

The total deflection of the beam may be presented in the following form:(21)$$w=\left(P_{b}+P_{s}\right)a$$

The angles of rotation of the face sheet beam can be expressed as:(22)$$\begin{array}{l} \psi =-\frac{\partial P_{b}}{\partial x}a=-\frac{2}{l_{e}}\frac{\partial P_{b}}{\partial \xi }a\\ \;=-\frac{2}{l_{e}}\left[\begin{array}{cccc} 0 & 1 & 2\xi & 3\xi^{2} \end{array}\right]a \end{array}$$

Substituting the values of the node coordinate, $\xi_{i}$ into Equation (21) and Equation (22), the nodal displacement of beam element is obtained as:(23)δ=Ca where $\delta^{T}=\left[\begin{array}{cccc} w_{1} & \psi_{1} & w_{2} & \psi_{2} \end{array}\right]$ and:(24)$$C=\frac{1}{l_{e}}\left[\begin{array}{cccc} l_{e} & -l_{e} & (1-2\alpha )l_{e} & (6\alpha -1)l_{e}\\ 0 & -2 & 4 & -6\\ l_{e} & l_{e} & (1-2\alpha )l_{e} & (1-6\alpha )l_{e}\\ 0 & -2 & -4 & -6 \end{array}\right]$$

The coefficient vector, a, can be determined from nodal displacement vector:(25)$$a=C^{-1}\delta$$

Substituting Equation (25) into Equation (16) and (18), the bending and shear deflection can be presented as follows:(26)$$w_{b}=P_{b}C^{-1}\delta$$

(27)$$w_{s}=P_{s}C^{-1}\delta$$

The total deflection of the beam may be expressed as:(28)$$w=\left(P_{b}+P_{s}\right)C^{-1}\delta =PC^{-1}\delta$$ where:(29)$$P=P_{b}+P_{s}$$

### 4.1. The Element Stiffness Matrix

The strain energy of bending deflection may be expressed as:(30)$$\Pi_{b}^{e}=\frac{1}{2}\int_{0}^{l_{e}}\int_{-h(x)/2}^{h(x)/2}\kappa_{b}^{T}\left[E(z){\left(z-h_{n}(x)\right)}^{2}\right]\kappa_{b}dzdx$$ where ${\left[\kappa \right]}_{b}$$\kappa_{b}$ is the bending curvature, which can be presented in the form:(31)$$\kappa_{b}=\left[-\frac{\partial^{2}w_{b}}{\partial x^{2}}\right]=-\frac{\partial^{2}P_{b}}{\partial x^{2}}C^{-1}\delta =-\frac{4}{l_{e}^{2}}\frac{\partial^{2}P_{b}}{\partial \xi^{2}}C^{-1}\delta$$

Given:(32)$$H_{b}=\frac{4}{l_{e}^{2}}\frac{\partial^{2}P_{b}}{\partial \xi^{2}}$$

(33)$$L_{b}=H_{b}C^{-1}$$

Then:(34)$$\kappa_{b}=-L_{b}\delta$$

By taking Equation (34) into account in Equation (30), the strain energy of bending is obtained as:(35)$$\Pi_{b}^{e}=\frac{1}{2}\frac{l_{e}}{2}\int_{-1}^{1}\int_{-h(\xi )/2}^{h(\xi )/2}\delta^{T}L_{b}^{T}\left[E(z){\left(z-h_{n}(\xi )\right)}^{2}\right]L_{b}\delta dzd\xi$$

The bending stiffness matrix of the beam element may be determined by:(36)$$K_{b}^{e}=\frac{l_{e}}{2}\int_{-1}^{1}\int_{-h(\xi )/2}^{h(\xi )/2}L_{b}^{T}\left[E(z){\left(z-h_{n}(\xi )\right)}^{2}\right]L_{b}dzd\xi$$

Substituting Equation (33) into Equation (36), yields:(37)$$K_{b}^{e}=C^{-T}B_{b}C^{-1}$$ where:(38)$$B_{b}=\frac{l_{e}}{2}\int_{-1}^{1}\int_{-h(\xi )/2}^{h(\xi )/2}H_{b}^{T}\left[E(z){\left(z-h_{n}(\xi )\right)}^{2}\right]H_{b}dzd\xi$$

The strain energy of shear deflection may be expressed as:(39)$$\Pi_{s}^{e}=\frac{1}{2}\int_{0}^{l_{e}}\int_{-h(x)/2}^{h(x)/2}\gamma^{T}\left[\frac{E(z)}{2\left(1+\nu \right)}\right]\gamma dzdx$$

The shear strain vector is obtained from Equation (3) and Equation (7):(40)$$\gamma =\left[\frac{\partial w_{s}}{\partial x}\right]=\frac{\partial P_{s}}{\partial x}C^{-1}\delta =\frac{2}{l_{e}}\frac{\partial P_{s}}{\partial \xi }C^{-1}\delta$$

Given:(41)$$H_{s}=\frac{2}{l_{e}}\frac{\partial P_{s}}{\partial \xi }$$

(42)$$L_{s}=H_{s}C^{-1}$$

The strain energy of shear deflection may be expressed as:(43)$$\Pi_{s}^{e}=\frac{1}{2}\frac{l_{e}}{2}\int_{-1}^{1}\int_{-h(\xi )/2}^{h(\xi )/2}\delta^{T}L_{s}^{T}\left[\frac{E(z)}{2\left(1+\nu \right)}\right]L_{s}\delta dzd\xi$$

The shear stiffness matrix of the beam element is obtained as:(44)$$K_{s}^{e}=\frac{l_{e}}{2}\int_{-1}^{1}\int_{-h(\xi )/2}^{h(\xi )/2}L_{s}^{T}\left[\frac{E(z)}{2\left(1+\nu \right)}\right]L_{s}dzd\xi$$

By substituting Equation (33) into Equation (36), yields:(45)$$K_{s}^{e}=C^{-T}B_{s}C^{-1}$$

In which:(46)$$B_{s}=\frac{l_{e}}{2}\int_{-1}^{1}\int_{-h(\xi )/2}^{h(\xi )/2}H_{s}^{T}\left[\frac{E(z)}{2\left(1+\nu \right)}\right]H_{s}dzd\xi$$

According to the account, the element stiffness matrix is:(47)$$K^{e}=K_{b}^{e}+K_{s}^{e}=C^{-T}\left(B_{b}+B_{s}\right)C^{-1}$$

Because $K_{s}^{e}$ depends on $\alpha =4D/Sl_{e}^{2}$, when the thickness of the beam is very small, then α≈0 and $K_{s}^{e}\approx 0$, and the transverse shear effects vanish, and, as a consequence, the proposed beam model is free of shear locking. For the functionally graded materials, the Poisson’s ratio is smaller than 0.4, so this model has no volume locking.

### 4.2. The Element Mass Matrix

The kinetic energy of beam element is obtained as:(48)$$U^{e}=\frac{1}{2}\int_{0}^{l_{e}}\int_{-h(x)/2}^{h(x)/2}\rho (z).{\dot{w}}^{2}dzdx=\frac{1}{2}\int_{0}^{l_{e}}\int_{-h(x)/2}^{h(x)/2}{\dot{\delta }}^{T}C^{-T}P^{T}\rho (z)PC^{-1}\dot{\delta }dzdx$$

According to Equation (48), the element mass matrix is obtained as:(49)$$M^{e}=\int_{0}^{l_{e}}\int_{-h(x)/2}^{h(x)/2}C^{-T}P^{T}\rho (z)PC^{-1}dzdx$$ or:(50)$$M^{e}=\frac{l_{e}}{2}\int_{-1}^{1}\int_{-h(\xi )/2}^{h(\xi )/2}C^{-T}P^{T}\rho (z)PC^{-1}dzd\xi$$

Given:(51)$$I_{0}=\frac{l_{e}}{2}\int_{-1}^{1}\int_{-h(\xi )/2}^{h(\xi )/2}P^{T}\rho (z)Pdzd\xi$$

The element mass matrix of the beam element is taken by:(52)$$M^{e}=C^{-T}I_{0}C^{-1}$$

### 4.3. The Element Geometric Stiffness Matrix

At the critical load, the beam takes the buckled form as shown in Figure 3.

According to Figure 3, the axial shortening of the beam can be taken as follows:(53)$$\begin{array}{l} ds=\sqrt{dx^{2}+dw^{2}}\\ \approx dx+\frac{1}{2}{\left(\frac{dw}{dx}\right)}^{2}dx\\ \Rightarrow du=\frac{1}{2}{\left(\frac{dw}{dx}\right)}^{2}dx \end{array}$$

The strain energy of an axial compressed load, Q, can be obtained as follows:(54)$$V^{e}=\frac{Q}{2}\int_{0}^{l}{\left(\frac{dw}{dx}\right)}^{2}dx=\frac{2}{l_{e}}\frac{Q}{2}\int_{-1}^{1}{\left(\frac{dw}{d\xi }\right)}^{2}d\xi$$

As a result, the element geometric stiffness matrix may be expressed as follows:(55)$$K_{g}^{e}=\frac{2}{l_{e}}\int_{-1}^{1}C^{-T}{\left[\frac{\partial P}{\partial \xi }\right]}^{T}Q\left[\frac{\partial P}{\partial \xi }\right]C^{-1}d\xi$$

Given:(56)$$H_{g}=\frac{2}{l_{e}}\frac{\partial^{2}P_{b}}{\partial \xi^{2}}$$

(57)$$G_{0}=\frac{2}{l_{e}}\int_{-1}^{1}H_{g}^{T}QH_{g}d\xi$$

Now, the element geometric stiffness matrix is hence given by:(58)$$K_{g}^{e}=C^{-T}G_{0}C^{-1}$$

### 4.4. The Element Load Vector

The work done by the transverse distribution load, q, is expressed as follows:(59)$$W^{e}=\int_{0}^{l_{e}}q\delta^{T}C^{-T}P^{T}dx$$

The element load vector may be obtained as:(60)$$F^{e}=\int_{0}^{l_{e}}qC^{-T}P^{T}dx=\frac{2}{l_{e}}\int_{-1}^{1}qC^{-T}P^{T}d\xi$$

### 4.5. Static Bending Solution

For static bending analysis, the nodal displacements can be obtained by solving the following equation:(61)Kδ=F where K, F are, respectively, the global stiffness matrix and global force vector of the beam.

### 4.6. Free Vibration Solution

The equation of motion for free vibration analysis of the beam is obtained as follows:(62)$$M\ddot{\delta }+K\delta =0$$ where M is the global mass matrix of the beam.

For free vibration analysis, assuming that:(63)$$\delta =\delta .e^{i\omega t}$$

By substituting Equation (63) into Equation (62), the natural frequency is obtained by solving the following eigenvalue equation:(64)(K−λM)δ=0 with $\sqrt{\lambda }=\omega$ denotes the natural frequency of the beam.

### 4.7. Buckling Solution

For buckling analysis and determination of the magnitude of a static compressive load that will produce beam buckling, the following eigenvalue equation will be achieved:(65)$$\left(K-Q_{cr}K_{g}\right)\delta =0$$ where $K_{g}$ is the global geometric stiffness matrix of the beam. 

The lowest positive eigenvalue of this equation is the magnitude of the critical buckling load, $Q_{cr}$, of the beam and the corresponding eigenvector is the deformed shape of the buckled beam.

## 5. Numerical Results and Discussion

### 5.1. Static Bending Analysis

#### 5.1.1. Verification

To verify the proposed beam model, in this section, the comparison of static deflection of an isotropic beam with variable thickness and an FGM beam with constant thickness is investigated.

Firstly, a simple-simple supported (S-S) isotropic beam with length, L, and variable thickness is considered here, the beam is under uniform distribution load, q, and the material properties of the beam are E=20.83 Msi,
G=3.71 Msi,
ν=0.44. The thickness of the beam varies along the *x*-direction by the following formula [36]:(66)$$h(x)=h_{0}\left[1+\lambda {\left(\frac{2x-L}{L}\right)}^{n}\right]$$ where $h_{0}$ is the height at the mid-span of the beam, λ is a small parameter, and n is the degree of non-uniformity. The non-dimensional mid-span deflection of the beam is calculated by the following formula [35]:(67)$$\bar{w}=\frac{100Eh_{0}^{3}}{12qL^{4}}w\left(\frac{L}{2}\right)$$

The comparison of non-dimensional mid-span deflection of an S-S supported isotropic beam using the proposed beam model with the results of Zenkour [35] and numerical results using Abaqus FE software packages (SIMULIA, Zaltbommel, Netherlands) are given in Table 1.

Secondly, an $Al/Al_{2}O_{3}$ beam with a constant thickness under a uniform distribution load, q, is investigated. The material properties of aluminum (as metal) and alumina (as ceramic) are [14]:


$\begin{array}{l} Al\;(\mathrm{metal}):\;E_{m}=70\;GPa,\;\nu_{m}=0.3,\;\rho_{m}=2702\;kg/m^{3}\\ Al_{2}O_{3}\;(\mathrm{ceramic}):\;E_{c}=380\;GPa,\;\nu_{c}=0.3,\;\rho_{c}=3960\;kg/m^{3} \end{array}$


The non-dimensional deflection at the central point of the beam for different values of the length-to-height ratio, L/h, is given by [14]:(68)$$\bar{w}=\frac{100E_{m}h^{3}}{qL^{4}}w\left(\frac{L}{2}\right)$$ where $E_{m}$ is the Young’s modulus of metal.

The comparison of non-dimensional deflection at the central point of the beam using the proposed beam model with the results of Thai et al. [14] (using the sinusoidal beam theory (SBT) and the hyperbolic beam theory (HBT)) and the results of Li et al. [20] (using analytical solutions) are listed in Table 2.

According to the two above comparisons, it can be observed that the values obtained using the proposed beam model are in good agreement with the published data.

#### 5.1.2. Numerical Results for Static Bending Analysis

In this section, we apply this model to explore the static bending response of a variable thickness FGM beam with length, L, the height at the left-hand end, $h_{0}$, subject to uniform distribution load, *q*. The boundary conditions of the beam are simple-simple supported (S-S) and clamped-clamped supported (C-C). 

For simple supported: w=0 at x=0, x=L.

For clamped supported: w=0, ψ=0 at x=0, x=L.

The Young’s modulus, mass density, and Poisson’s ratio are:

Al (metal): $E_{m}=70\;GPa,$$\rho_{m}=2702\;kg/m^{3},$$\nu_{m}=0.3$.

$Al_{2}O_{3}\;(\mathrm{ceramic})$: $E_{c}=380\;GPa,$$\rho_{c}=3960\;kg/m^{3},$$\nu_{c}=0.3$.

The height of the beam varies along the x-direction as the following formula:(69)$$h(x)=\frac{h_{0}}{2}\left[1+{\left(\frac{L-x}{L}\right)}^{n}\right]$$ where n is the degree of non-uniformity.

The maximum non-dimensional transverse deflection of the beam is expressed by:(70)$${\bar{w}}_{\max}=\frac{100E_{m}h_{0}^{3}}{qL^{4}}w_{\max}$$

The maximum non-dimensional transverse deflection of the beam with different values of index, p=0, 0.5, 1, 2, 5, 10, index n=0,0.5, 1, 2 (n=0 corresponding to the constant thickness beam), and the length-to-height ratio, $L/h_{0}=10,\;20,\;50,\;100$, is listed in Table 3, Figure 4, Figure 5, Figure 6 and Figure 7.

To illustrate the effects of the power-law index, p, and the degree of non-uniformity, n, on the bending response of a variable thickness FGM beam subjected to uniform load, the non-dimensional transverse deflections of the beam are given in Table 3 and plotted in Figure 4, Figure 5, Figure 6 and Figure 7.

According to Table 3 and Figure 4, Figure 5, Figure 6 and Figure 7, it shows that the maximum non-dimensional transverse deflection of the beam increases as a function of the power-law index, p. It means that the richer metal FGM beam is more flexible than the richer ceramic FGM beam. The maximum non-dimensional transverse deflection of the beam increases rapidly when the power-law index, p, increases in the range of 0÷1.

The influence of index n on the maximum non-dimensional transverse deflection of the beam is shown in Table 3, Figure 4, Figure 5, Figure 6 and Figure 7. By increasing index n, the maximum non-dimensional transverse deflection of the beam increases. When index n increases in the range of 0÷1, the transverse deflection of the beam increases rapidly. When n=0 and n=∞, the heights of the beam are constant, $h(x)=h_{0}$ and $h(x)=h_{0}/2$, so that the maximum deflection position appears at the mid-span of the beam. 

In addition, when the length-to-height ratio, $L/h_{0}$, increases, the transverse deflection of the beam increases. The deflection affected by the boundary condition on the static bending response of the beam is shown in Figure 4 and Figure 5. In general, the deformation shape of the variable thickness beam is not symmetric. The transverse deflection of the C-C supported beam is smaller than the one of the S-S supported beam.

Figure 8 shows the distribution of the axial normal stress, $\sigma_{x}$, and shear stress, $\tau_{xy}$, across the z-direction at the mid-span. It shows that when p=0, the axial normal stress, $\sigma_{x}$, has linear distribution while the shear stress, $\tau_{xz}$, is constant along the height of the beam.

### 5.2. Free Vibration Analysis

#### 5.2.1. Verification

To confirm the accuracy of the proposed beam model, a comparison of frequencies of a cantilever isotropic tapered beam with length, L and different values of the taper ratio, $c=1-h_{1}/h_{0}$, is considered herein. The non-dimensional natural frequency of the beam is defined as [34]:(71)$${\bar{\omega }}_{i}=\omega_{i}\sqrt{\frac{m_{0}L^{4}}{EI_{0}}}$$ where $m_{0},\;I_{0}$ are the mass per unit length and the flexural rigidity at the left-hand end of the beam, respectively. The comparison of non-dimensional natural frequencies using the proposed beam model with the results of Banerjee et al. [34] and the numerical results using Abaqus FE software packages is shown in Table 4.

A (S-S) $Al/Al_{2}O_{3}$ beam with constant thickness and different values of the ratio of L/h is considered. The material properties of Al (metal) and $Al_{2}O_{3}$ (ceramic) are [14]:


Al (metal): Em=70 GPa, νm=0.3, ρm=2702 kg/m3Al2O3 (ceramic): Ec=380 GPa, νc=0.3, ρc=3960 kg/m3.


The non-dimensional natural frequencies of the beam are defined as [14]:(72)$${\bar{\omega }}_{i}=\omega_{i}\frac{L^{2}}{h}\sqrt{\frac{\rho_{m}}{E_{m}}}$$

Table 5 shows the first three non-dimensional natural frequencies of the beam using the proposed beam model and the results of Thai et al. [14] (using SBT and HBT).

The present frequencies are in good agreement with the other published results.

#### 5.2.2. Numerical Results of Free Vibration Analysis

In this section, the variable thickness beam, which is given in the static bending problem, is employed again herein. The non-dimensional frequencies are obtained as:(73)$${\bar{\omega }}_{i}=\omega_{i}\frac{L^{2}}{h_{0}}\sqrt{\frac{\rho_{m}}{E_{m}}}$$

The results of the free vibration analysis of the variable thickness FGM beam with different values of the length-to-height ratio, $L/h_{0}$, power-law index, p, index n, and boundary conditions are reported in Table 6 and Table 7, and Figure 9, Figure 10, Figure 11 and Figure 12.

By considering Table 6 and Table 7, the non-dimensional frequencies of the C-C beam are higher than the ones of the S-S beam. Furthermore, the non-dimensional frequencies decrease as a function of the length-to-height ratio, $L/h_{0}$.

The influences of index p and the degree of non-uniformly, n, on the free vibration of the beam are considered here. Table 6 and Figure 9 show that the index, p, has a strong effect on the non-dimensional frequencies of the beam. It can be seen that increasing the index, p, will reduce the stiffness of the FGM beam, which leads to a decrease in the non-dimensional frequencies of the beam. This is due to the fact that higher values of index p correspond to a high portion of the metal in comparison with the ceramic portion, and, consequently, the FGM beam becomes more flexible. According to Table 6 and Table 7, and Figure 10, when increasing the index, n, the non-dimensional frequencies of the beam decrease. In fact, the fundamental non-dimensional frequency of the beam decreases rapidly when the index, n, increases in the range of 0÷2. Figure 11 plots the first four mode shapes of the S-S support variable thickness FGM beam with $L/h_{0}=10$, p=0.5, and n=2. Figure 12 plots the first mode shapes of the variable thickness FGM beam with $L/h_{0}=10,\;p=0.5$, and n=2 for some boundary conditions. It can be seen that the mode shapes of the variable thickness FGM beam show greater differences from those of the constant thickness FGM beam. The amplitude of vibration at the slender end is higher than that at the other end.

### 5.3. Buckling Analysis

#### 5.3.1. Verification

To verify the proposed beam model for buckling analysis, S-S and C-C isotropic beams with a constant thickness are considered. The length of the beam is *L* = 1, and the modulus of elasticity and Poisson’s ratio of material are, respectively, $E=10^{3}\;GPa,$ν=0.333. ${\bar{Q}}_{cr}=Q_{cr}\frac{12L^{2}}{Eh_{}^{3}}$

The present buckling load results are compared with the analytical solutions (which are given in [8]), the results of Ferreira [52] and Oreh et al. [8], and numerical results using Abaqus FE software packages and are finally reported in Table 8.

Furthermore, an $Al/Al_{2}O_{3}$ beam with a constant thickness subjected to a uniaxial load is investigated here. The material properties of the two components of the FGM beam are [9]:


Al (metal): Em=70 GPa, νm=0.3, ρm=2702 kg/m3Al2O3 (ceramic): Ec=380 GPa, νc=0.3, ρc=3960 kg/m3.


The length-to-height ratio is L/h=5 and L/h=10, and the boundary condition of the beam is S-S supported. The non-dimensional buckling load is defined as [9]:(74)$${\bar{Q}}_{cr}=Q_{cr}\frac{12L^{2}}{E_{m}h_{}^{3}}$$

Table 9 shows the comparison between the present non-dimensional buckling load and the results of Li et al. [9] using the analytical solution.

The two comparisons indicate a good agreement between the present results by using this model and published results.

#### 5.3.2. Numerical Results for Buckling Analysis

The variable thickness FGM beam, which is given in the static bending problem, is again considered herein. The beam is subject to a uniaxial compressed load, Q.

The non-dimensional critical load is obtained by:(75)$${\bar{Q}}_{cr}=Q_{cr}\frac{12L^{2}}{E_{m}h_{0}^{3}}$$

The effect of the power-law index, p, on the critical load of the variable thickness FGM beam is given in Table 10 and Figure 13. It implies that the power-law index, p, strongly affects the critical buckling load of the FGM beam in some range. When the power-law index, p, is increasing in the range of 0÷2, the critical load decreases rapidly, while p>2, the index, p, has a slight effect.

In Table 10 and Figure 14, the critical load is a function of the degree of non-uniformity, n. In general, the critical load decreases when increasing index n. In addition, we can see again in Table 10 that the critical load of the C-C supported variable thickness beam is much higher than the S-S supported beam.

Figure 15 and Figure 16 show the first buckling mode shapes of variable thickness FGM beam with $p=0.5,\;n=0.5,\;L/h_{0}=10$ and $p=0.5,\;n=2,\;L/h_{0}=10$ for some boundary conditions.

## 6. Conclusions

The paper established the modified first order shear deformation beam theory, and applied this theory to model the variable thickness FGM beam element. Using the proposed beam model, the static bending, free vibration, and buckling of the variable thickness FGM beam were investigated and simulated. This beam model can effectively simulate the mechanical behavior of FGM beams by comparing it with results of other methods and the numerical simulations using Abaqus FE software packages. Parameters that study the influence of geometry, materials, and boundary conditions on the mechanical behavior of the beam were considered.

The advantage of the proposed beam model is its simplicity in formulation, and high convergence and free shear-locking without reduced integration or selective reduced integration. This proposed beam model can be applied widely to simulate the mechanical behavior of FGM beams. 

## Figures and Tables

**Figure 1 materials-12-00404-f001:**
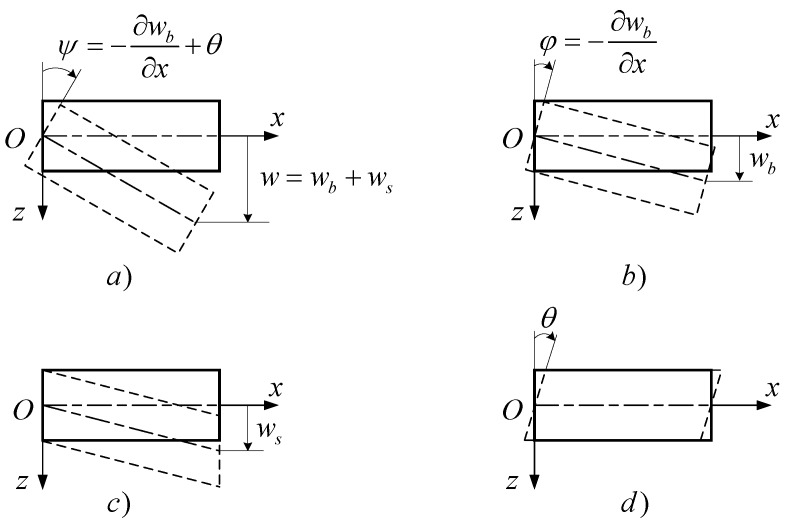
The beam model with bending and shear deflection. (**a**) Total deflection and rotation, (**b**) pure bending and rotation, (**c**) transverse shear deflection, (**d**) shear angle rotation.

**Figure 2 materials-12-00404-f002:**
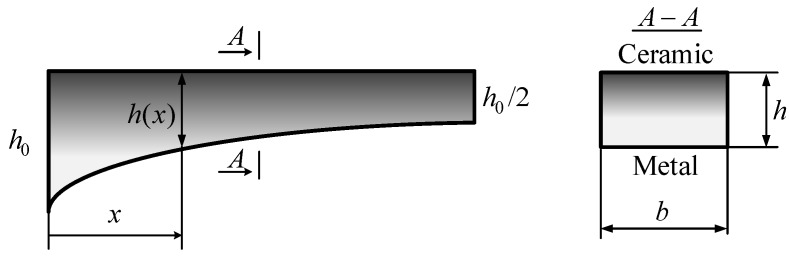
Modeling of variable thickness FGM beam.

**Figure 3 materials-12-00404-f003:**
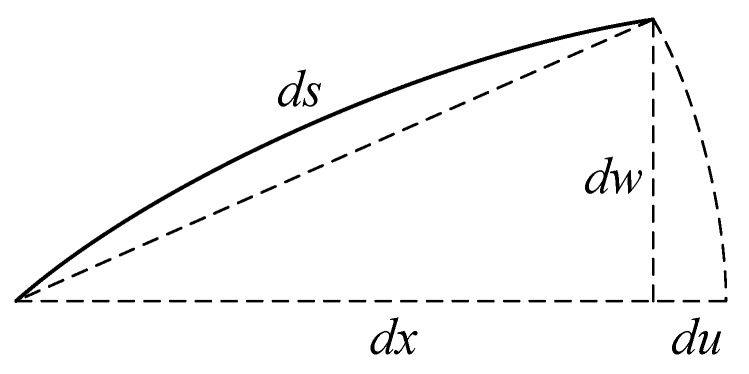
The buckled form of the beam.

**Figure 4 materials-12-00404-f004:**
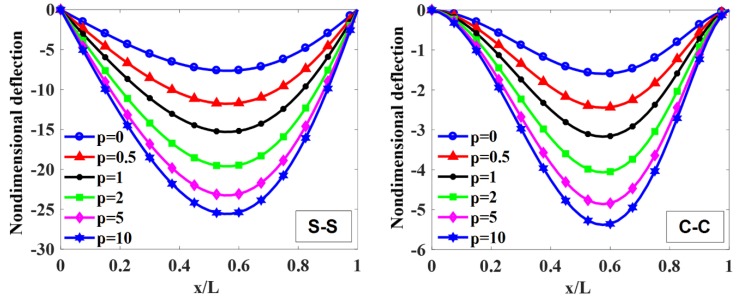
The non-dimensional transverse deflection of S-S and C-C supported FGM beam with different values of index p when $L/h_{0}=10$ and n=1.

**Figure 5 materials-12-00404-f005:**
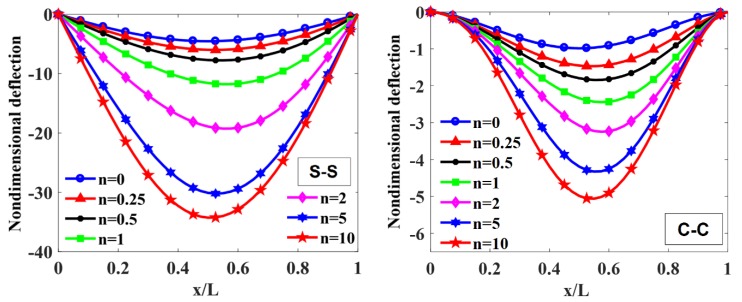
The non-dimensional transverse deflection of S-S and C-C supported FGM beam with different values of index n with $L/h_{0}=10$ and p=0.5.

**Figure 6 materials-12-00404-f006:**
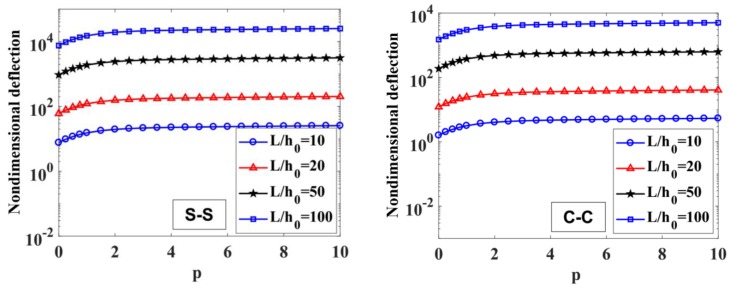
The maximum non-dimensional transverse deflection, ${\bar{w}}_{\max}$, of S-S and C-C supported FGM beam depend on index p with $L/h_{0}=10,\;20,\;50,\text{​ }100$ and n=1.

**Figure 7 materials-12-00404-f007:**
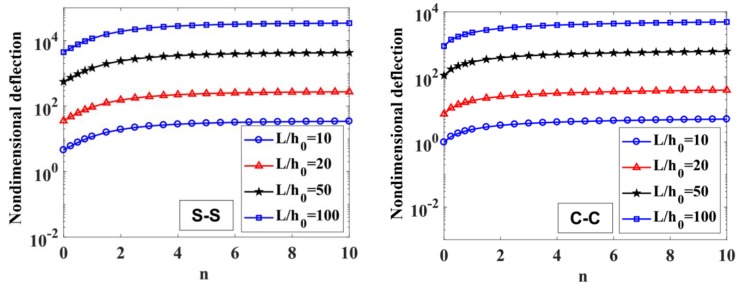
The maximum non-dimensional transverse deflection, ${\bar{w}}_{\max}$, of S-S and C-C supported FGM beam depending on index n with $L/h_{0}=10,\;20,\;50,\text{​ }100$ and p=0.5.

**Figure 8 materials-12-00404-f008:**
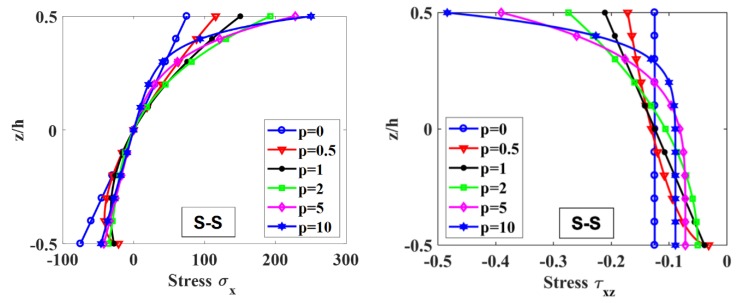
Distribution of axial normal stress, $\sigma_{x}$, and shear stress, $\tau_{xz}$, at the mid-span of the beam across the height of S-S supported FGM beam under uniform load with $L/h_{0}=\;100$.

**Figure 9 materials-12-00404-f009:**
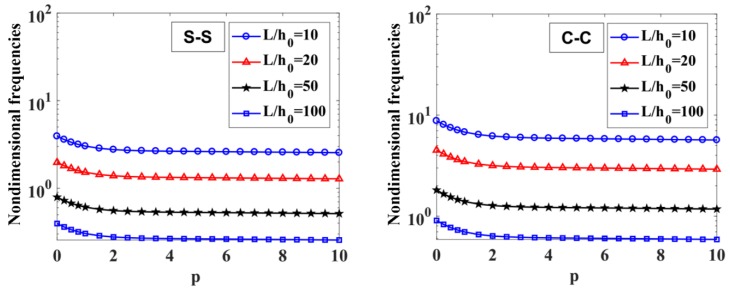
The influence of index p on the fundamental non-dimensional natural frequencies, ${\bar{\omega }}_{1}$, of S-S and C-C supported FGM beam with $L/h_{0}=10,\;20,\;50,\text{​ }100$ and n=1.

**Figure 10 materials-12-00404-f010:**
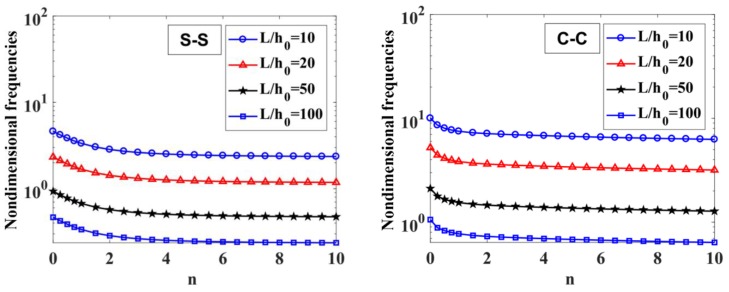
The influence of n on the fundamental non-dimensional natural frequencies, ${\bar{\omega }}_{1}$, of S-S and C-C supported FGM beam with $L/h_{0}=10,\;20,\;50,\text{​ }100$ and p=0.5.

**Figure 11 materials-12-00404-f011:**
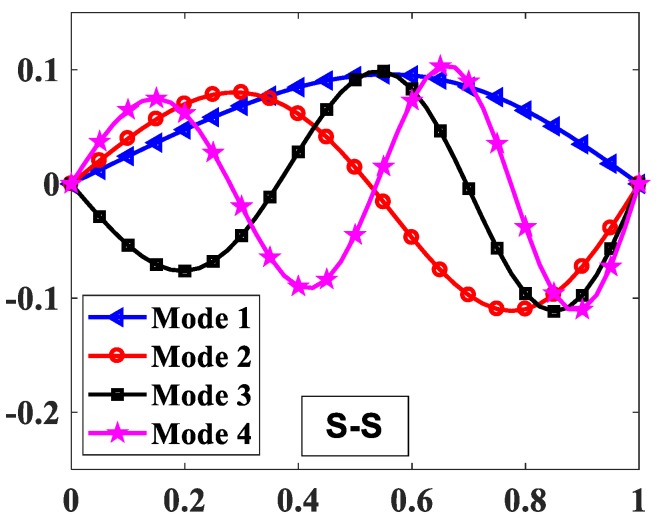
The first four mode shapes of the FGM beam with $p=0.5,\;n=2,\;L/h_{0}=10$.

**Figure 12 materials-12-00404-f012:**
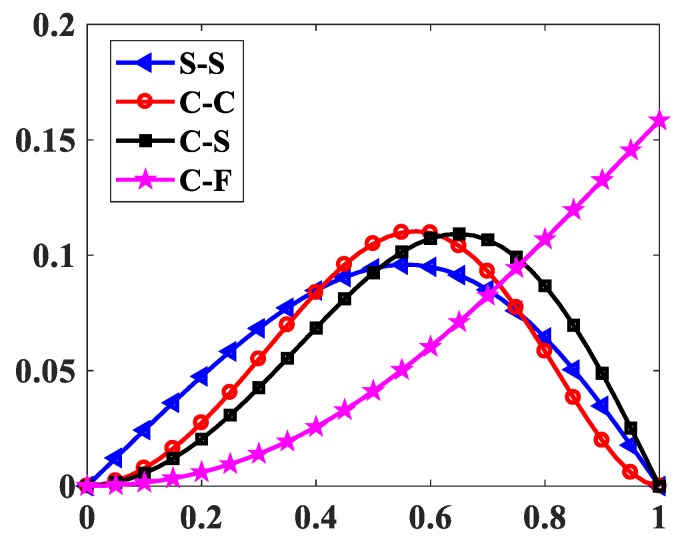
The first mode shapes of the FGM beam with $p=0.5,\;n=2,\;L/h_{0}=10$.

**Figure 13 materials-12-00404-f013:**
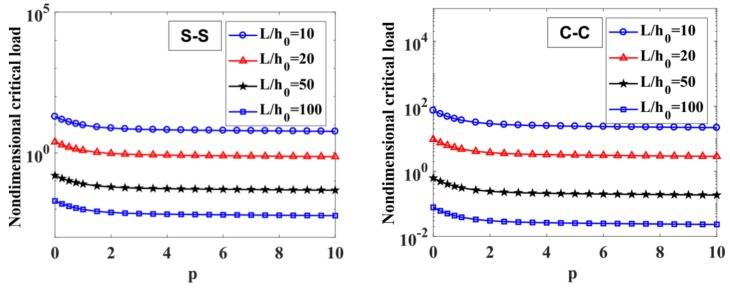
The influence of index p on the non-dimensional critical load, ${\bar{Q}}_{cr}$, of an S-S and C-C supported FGM beam with $L/h_{0}=10,\;20,\;50,\text{​ }100$ and n=1.

**Figure 14 materials-12-00404-f014:**
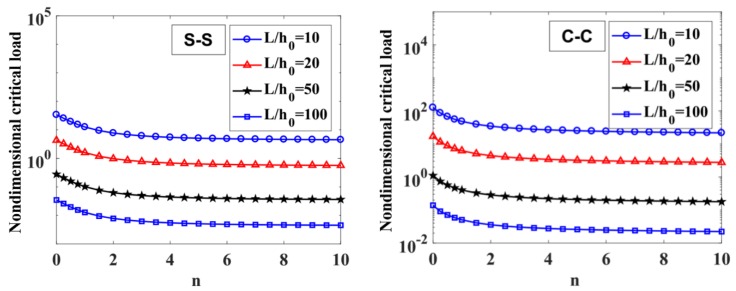
The influence of index n on the non-dimensional critical load, ${\bar{Q}}_{cr}$, of S-S and C-C supported FGM beam with $L/h_{0}=10,\;20,\;50,\text{​ }100$ and p=0.5.

**Figure 15 materials-12-00404-f015:**
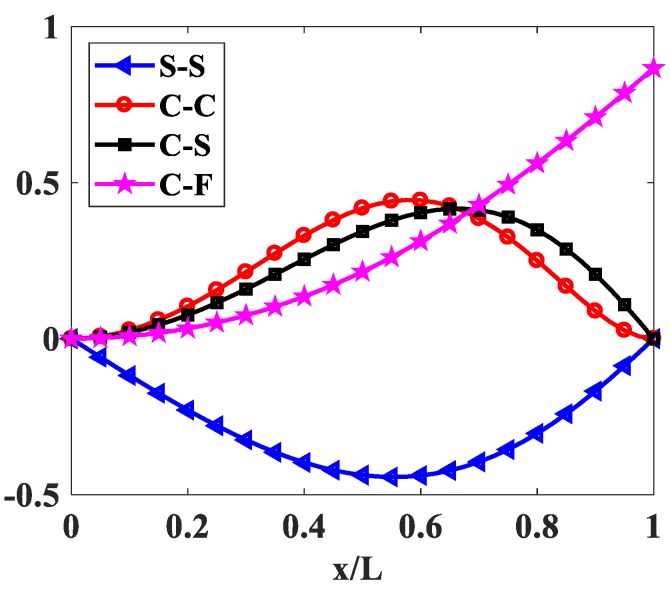
The first buckling mode shapes of the FGM beam with $p=0.5,\;n=0.5,\;L/h_{0}=10$.

**Figure 16 materials-12-00404-f016:**
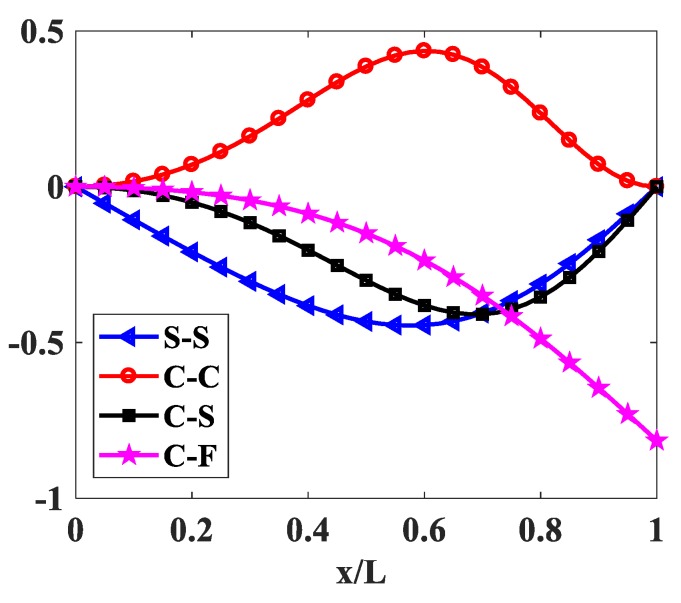
The first buckling mode shapes of the FGM beam with $p=0.5,\;n=2,\;L/h_{0}=10$.

**Table 1 materials-12-00404-t001:** Comparison of non-dimensional mid-span deflection of an S-S supported isotropic beam for different values of the thickness parameter, λ, n with $L/h_{0}=10$.

λ	n = 1			n = 2			n = 3		
	[35]	Abaqus	Present	[35]	Abaqus	Present	[35]	Abaqus	Present
0.05	1.3370	1.3500	1.3405	1.3111	1.3237	1.3144	1.3349	1.3356	1.3385
0.10	1.3441	1.3584	1.3493	1.2889	1.3021	1.2920	1.3359	1.3368	1.3395
0.20	1.3727	1.3967	1.3849	1.2479	1.2626	1.2508	1.3400	1.3416	1.3437
0.30	1.4204	1.4625	1.4497	1.2103	1.2279	1.2138	1.3467	1.3500	1.3510
0.40	1.4872	1.5534	1.5502	1.1747	1.1956	1.1801	1.3561	1.3428	1.3632

**Table 2 materials-12-00404-t002:** Comparison of non-dimensional deflection at the central point of the S-S supported FGM beam with constant thickness and different values of the ratio of L/h=5,, L/h=20.

*p*	*L*/*h* = 5				*L*/*h* = 20			
	[20]	SBT [35]	HBT [35]	Present	[20]	SBT [35]	HBT [35]	Present
0	3.1657	3.1649	3.1654	3.1657	2.8962	2.8962	2.8962	2.8962
0.5	4.8292	4.8278	4.8285	4.8348	4.4645	4.4644	4.4644	4.4648
1	6.2599	6.2586	6.2594	6.2599	5.8049	5.8049	5.8049	5.8049
2	8.0602	8.0683	8.0675	8.0303	7.4415	7.4421	7.4420	7.4397
5	9.7802	9.8367	9.8271	9.6483	8.8151	8.8188	8.8181	8.8069
10	10.8979	10.9420	10.9375	10.7194	9.6879	9.6908	9.6905	9.6767

**Table 3 materials-12-00404-t003:** Maximum non-dimensional transverse deflection, ${\bar{w}}_{\max}$, of a variable thickness beam.

$L/h_{0}$	p	S-S				C-C			
		n = 0	n = 0.5	n = 1	n = 2	n = 0	n = 0.5	n = 1	n = 2
10	0	2.9501	5.0428	7.6450	12.4967	0.6475	1.2046	1.5965	2.1175
	0.5	4.5388	7.7639	11.7761	19.2590	0.9867	1.8432	2.4462	3.2483
	1	5.8959	10.0885	15.3053	25.0363	1.2763	2.3884	3.1717	4.2139
	2	7.5578	12.9313	19.6172	32.0881	1.6376	3.0634	4.0674	5.4034
	5	8.9752	15.3402	23.2520	38.0049	1.9745	3.6699	4.8622	6.4472
	10	9.8853	16.8831	25.5737	41.7764	2.1995	4.0688	5.3818	7.1264
20	0	2.8962	4.9748	7.5688	12.4132	0.5936	1.1383	1.5228	2.0363
	0.5	4.4648	7.6707	11.6719	19.1450	0.9127	1.7523	2.3451	3.1368
	1	5.8049	9.9740	15.1774	24.8963	1.1853	2.2770	3.0475	4.0769
	2	7.4397	12.7829	19.4513	31.9066	1.5194	2.9186	3.9062	5.2256
	5	8.8069	15.1288	23.0158	37.7459	1.8063	3.4628	4.6323	6.1939
	10	9.6767	16.6219	25.2822	41.4567	1.9910	3.8118	5.0969	6.8128
50	0	2.8812	4.9578	7.5456	12.4021	0.5785	1.1233	1.5022	2.0150
	0.5	4.4441	7.6483	11.6397	19.1344	0.8920	1.7328	2.3168	3.1082
	1	5.7794	9.9470	15.1372	24.8863	1.1598	2.2536	3.0127	4.0421
	2	7.4066	12.7482	19.3989	31.8952	1.4864	2.8883	3.8610	5.1806
	5	8.7597	15.0791	22.9408	37.7255	1.7591	3.4164	4.5675	6.1296
	10	9.6183	16.5627	25.1875	41.4396	1.9326	3.7526	5.0161	6.7344
100	0	2.8790	4.9590	7.5518	12.4096	0.5764	1.1247	1.5013	2.0129
	0.5	4.4413	7.6516	11.6519	19.1490	0.8890	1.7356	2.3162	3.1058
	1	5.7758	9.9518	15.1541	24.9056	1.1561	2.2575	3.0123	4.0393
	2	7.4018	12.7551	19.4218	31.9211	1.4816	2.8933	3.8607	5.1771
	5	8.7529	15.0879	22.9710	37.7591	1.7524	3.4219	4.5666	6.1242
	10	9.6098	16.5766	25.2311	41.4862	1.9242	3.7590	5.0162	6.7286

**Table 4 materials-12-00404-t004:** Comparison of the first three non-dimensional natural frequencies for a cantilever isotropic bean with different values of the taper ratio.

*c*	${\bar{\omega }}_{1}$			${\bar{\omega }}_{2}$			${\bar{\omega }}_{3}$		
	[34]	Abaqus	Present	[34]	Abaqus	Present	[34]	Abaqus	Present
0.1	3.559	3.562	3.553	21.338	21.140	21.132	58.980	57.510	57.663
0.2	3.608	3.612	3.603	20.621	20.453	20.439	56.192	54.939	55.045
0.3	3.667	3.669	3.662	19.881	19.739	19.720	53.322	52.269	52.331
0.4	3.737	3.739	3.732	19.114	18.996	18.975	50.354	49.487	49.513
0.5	3.824	3.826	3.819	18.317	18.222	18.198	47.265	46.568	46.565
0.6	3.934	3.936	3.930	17.488	17.413	17.391	44.025	43.482	43.464
0.7	4.082	4.083	4.078	16.625	16.568	16.548	40.588	40.186	40.155
0.8	4.292	4.293	4.290	15.743	15.701	15.691	36.885	36.608	36.583
0.9	4.631	4.631	4.630	14.931	14.902	14.911	32.833	32.671	32.688

**Table 5 materials-12-00404-t005:** Comparison of the first three non-dimensional natural frequencies of $Al/Al_{2}O_{3}$ beam with L/h=5 and L/h=20.

*L/h*	*p*	${\bar{\omega }}_{1}$			${\bar{\omega }}_{2}$			${\bar{\omega }}_{3}$		
		SBT [14]	HBT [14]	Present	SBT [14]	HBT [14]	Present	SBT [14]	HBT [14]	Present
5	0	5.1531	5.1527	5.2220	17.8868	17.8810	18.4730	34.2344	34.2085	35.6198
	0.5	4.4110	4.4107	4.4693	15.4631	15.4587	15.9861	29.8569	29.8373	31.1588
	1	3.9907	3.9904	4.0497	14.0138	14.0098	14.5588	27.1152	27.0971	28.5214
	2	3.6263	3.6265	3.6936	12.6411	12.6407	13.2636	24.3237	24.3151	25.9539
	5	3.3998	3.4014	3.4882	11.5324	11.5444	12.3067	21.6943	21.7187	23.6695
	10	3.2811	3.2817	3.3644	11.0216	11.0246	11.7210	20.5581	20.5569	22.2828
20	0	5.4603	5.4603	5.4658	21.5736	21.5732	21.6578	47.5950	47.5930	47.9905
	0.5	4.6511	4.6511	4.6556	18.3965	18.3962	18.4665	40.6542	40.6526	40.9852
	1	4.2051	4.2051	4.2096	16.6347	16.6344	16.7048	36.7692	36.7679	37.1020
	2	3.8361	3.8361	3.8413	15.1617	15.1619	15.2418	33.4681	33.4691	33.8471
	5	3.6484	3.6485	3.6554	14.3728	14.3748	14.4806	31.5699	31.5789	32.0740
	10	3.5389	3.5390	3.5457	13.9255	13.9264	14.0289	30.5337	30.5373	31.0136

**Table 6 materials-12-00404-t006:** First three non-dimensional natural frequencies, ${\bar{\omega }}_{i},\;i=1,2,3$, of S-S and C-C supported FGM beam with $L/h_{0}=10$ depending on index p and index n.

Mode	p	S-S				C-C			
		n = 0	n = 0.5	n = 1	n = 2	n = 0	n = 0.5	n = 1	n = 2
1	0	5.4144	4.5186	3.9296	3.3404	11.6989	9.3900	8.7793	8.3438
	0.5	4.6165	3.8514	3.3485	2.8457	10.0244	8.0292	7.5020	7.1257
	1	4.1761	3.4834	3.0282	2.5733	9.0882	7.2725	6.7930	6.4505
	2	3.8104	3.1785	2.7632	2.3481	8.2881	6.6336	6.1967	5.8846
	5	3.6201	3.0214	2.6277	2.2339	7.8125	6.2736	5.8667	5.5765
	10	3.5072	2.9282	2.5476	2.1664	7.5246	6.0571	5.6688	5.3921
2	0	20.8896	17.5473	15.6625	13.8003	30.2375	25.1511	23.2809	21.5200
	0.5	17.8784	14.9960	13.3746	11.7763	26.0655	21.6021	19.9692	18.4359
	1	16.1999	13.5793	12.1067	10.6567	23.6957	19.6054	18.1126	16.7125
	2	14.7756	12.3871	11.0448	9.7226	21.5964	17.8751	16.5163	15.2415
	5	13.9541	11.7253	10.4682	9.2250	20.1629	16.7856	15.5430	14.3714
	10	13.4588	11.3281	10.1235	8.9283	19.2868	16.1225	14.9524	13.8450
3	0	44.4998	37.8226	34.0651	30.3772	55.1307	47.1283	43.6107	40.0379
	0.5	38.2849	32.4491	29.1820	25.9891	47.8038	40.6706	37.5628	34.4232
	1	34.7730	29.4349	26.4536	23.5455	43.5761	36.9917	34.1349	31.2560
	2	31.6987	26.8401	24.1253	21.4760	39.6909	33.7104	31.1133	28.4944
	5	29.6875	25.2494	22.7497	20.2930	36.7108	31.4169	29.0861	26.7146
	10	28.4609	24.2835	21.9179	19.5804	34.8885	30.0141	27.8481	25.6299

**Table 7 materials-12-00404-t007:** Fundamental non-dimensional natural frequencies, ${\bar{\omega }}_{1}$ of (S-S), and (C-C) FGM beams depend on ratio, $L/h_{0}$, and index n.

$L/h_{0}$	p	S-S				C-C			
		n = 0	n = 0.5	n = 1	n = 2	n = 0	n = 0.5	n = 1	n = 2
10	0	5.4144	4.5186	3.9296	3.3404	11.6989	9.3900	8.7793	8.3438
	0.5	4.6165	3.8514	3.3485	2.8457	10.0244	8.0292	7.5020	7.1257
	1	4.1761	3.4834	3.0282	2.5733	9.0882	7.2725	6.7930	6.4505
	2	3.8104	3.1785	2.7632	2.3481	8.2881	6.6336	6.1967	5.8846
	5	3.6201	3.0214	2.6277	2.2339	7.8125	6.2736	5.8667	5.5765
	10	3.5072	2.9282	2.5476	2.1664	7.5246	6.0571	5.6688	5.3921
20	0	2.7329	2.2752	1.9750	1.6760	6.1175	4.8350	4.4997	4.2589
	0.5	2.3278	1.9377	1.6820	1.4273	5.2178	4.1212	3.8349	3.6291
	1	2.1048	1.7520	1.5207	1.2904	4.7208	3.7276	3.4683	3.2820
	2	1.9207	1.5987	1.3877	1.1775	4.3072	3.4012	3.1647	2.9947
	5	1.8277	1.5215	1.3208	1.1209	4.0899	3.2328	3.0088	2.8478
	10	1.7729	1.4759	1.2814	1.0875	3.9606	3.1328	2.9163	2.7608
50	0	1.0961	0.9118	0.7912	0.6708	2.4797	1.9477	1.8127	1.7133
	0.5	0.9334	0.7763	0.6737	0.5712	2.1120	1.6585	1.5437	1.4589
	1	0.8438	0.7019	0.6091	0.5164	1.9096	1.4993	1.3957	1.3190
	2	0.7700	0.6405	0.5558	0.4712	1.7425	1.3682	1.2736	1.2036
	5	0.7331	0.6097	0.5292	0.4486	1.6584	1.3025	1.2124	1.1457
	10	0.7114	0.5915	0.5135	0.4352	1.6087	1.2636	1.1763	1.1114
100	0	0.5483	0.4559	0.3955	0.3353	1.2422	0.9734	0.9069	0.8572
	0.5	0.4668	0.3881	0.3368	0.2855	1.0578	0.8287	0.7721	0.7299
	1	0.4221	0.3509	0.3044	0.2581	0.9563	0.7492	0.6981	0.6598
	2	0.3851	0.3202	0.2778	0.2355	0.8727	0.6837	0.6370	0.6021
	5	0.3667	0.3048	0.2645	0.2242	0.8308	0.6509	0.6064	0.5732
	10	0.3558	0.2957	0.2566	0.2175	0.8062	0.6314	0.5883	0.5560

**Table 8 materials-12-00404-t008:** Comparison of the buckling load of the constant thickness beam with different values of the ratio of L/h for S-S and C-C support conditions.

L/h	S-S					C-C				
	Analytical Solution [8]	[52]	[8]	Abaqus	Present	Analytical Solution [8]	[52]	[8]	Abaqus	Present
10	8013.8	8021.8	8013.86	8020.9	8013.83	29766	29877	29770	29864	29767.2
100	8.223	8.231	8.2225	8.2258	8.2225	32.864	32.999	32.864	32.917	32.864
1000	0.0082	0.0082	0.00822	0.00823	0.00822	0.0329	0.0330	0.0329	0.03295	0.0329

**Table 9 materials-12-00404-t009:** Comparison of the nondimensional buckling load of the constant thickness FGM beam with the ratio of L/h=5 and L/h=10 for the S-S and C-C support condition.

*p*	L/h = 5, C-C		L/h = 5, S-S		L/h = 10, C-C		L/h = 10, S-S	
	[9]	Present	[9]	Present	[9]	Present	[9]	Present
0	154.35	154.37	48.835	48.836	195.34	195.35	52.309	52.308
0.5	103.22	103.23	31.967	31.968	127.87	127.87	33.996	33.997
1	80.498	80.505	24.687	24.687	98.749	98.752	26.171	26.171
2	62.614	62.620	19.245	19.245	76.980	76.983	20.416	20.416
5	50.384	50.389	16.024	16.024	64.096	64.099	17.192	17.194
10	44.267	44.272	14.427	14.427	57.708	57.711	15.612	15.612

**Table 10 materials-12-00404-t010:** Non-dimensional critical load, ${\bar{Q}}_{cr}$, of an S-S and C-C variable thickness FGM beam depending on the ratio of $L/h_{0}$ and index n.

$L/h_{0}$	p	S-S				C-C			
		n = 0	n = 0.5	n = 1	n = 2	n = 0	n = 0.5	n = 1	n = 2
10	0	52.2381	29.9212	19.4184	11.9650	194.3954	103.3309	73.8638	52.8577
	0.5	33.9560	19.4365	12.6075	7.7642	127.3125	67.4450	48.1566	34.4249
	1	26.1410	14.9587	9.7007	5.9727	98.3404	52.0179	37.1226	26.5247
	2	20.3927	11.6700	7.5685	4.6601	76.6582	40.5622	28.9508	20.6879
	5	17.1703	9.8358	6.3844	3.9343	63.7783	33.9271	24.2601	17.3648
	10	15.5882	8.9358	5.8042	3.5789	57.3891	30.6474	21.9459	15.7276
20	0	6.6546	3.7940	2.4532	1.5062	26.1201	13.5473	9.6064	6.8244
	0.5	4.3168	2.4607	1.5908	0.9766	16.9785	8.7970	6.2363	4.4291
	1	3.3203	1.8925	1.2234	0.7510	13.0709	6.7692	4.7982	3.4074
	2	2.5907	1.4766	0.9546	0.5860	10.1967	5.2811	3.7435	2.6585
	5	2.1884	1.2476	0.8067	0.4953	8.5855	4.4534	3.1582	2.2438
	10	1.9917	1.1355	0.7344	0.4510	7.7944	4.0473	2.8713	2.0406
50	0	0.4282	0.2437	0.1575	0.0965	1.7075	0.8770	0.6219	0.4404
	0.5	0.2776	0.1580	0.1021	0.0625	1.1074	0.5685	0.4032	0.2855
	1	0.2135	0.1215	0.0785	0.0481	0.8516	0.4371	0.3101	0.2195
	2	0.1666	0.0948	0.0613	0.0375	0.6645	0.3411	0.2419	0.1713
	5	0.1408	0.0801	0.0518	0.0317	0.5616	0.2883	0.2045	0.1448
	10	0.1283	0.0730	0.0472	0.0289	0.5112	0.2625	0.1863	0.1318
100	0	0.0536	0.0305	0.0197	0.0121	0.2141	0.1095	0.0778	0.0551
	0.5	0.0347	0.0197	0.0128	0.0078	0.1388	0.0710	0.0504	0.0357
	1	0.0267	0.0152	0.0098	0.0060	0.1067	0.0546	0.0387	0.0274
	2	0.0208	0.0118	0.0076	0.0047	0.0833	0.0426	0.0302	0.0214
	5	0.0176	0.0100	0.0065	0.0040	0.0704	0.0360	0.0256	0.0181
	10	0.0160	0.0091	0.0059	0.0036	0.0641	0.0328	0.0233	0.0165

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
