# Peer review of "A New Beam Model for Simulation of the Mechanical Behaviour of Variable Thickness Functionally Graded Material Beams Based on Modified First Order Shear Deformation Theory"

_materials, 2019, doi:10.3390/ma12030404_

Round 1
Reviewer 1 Report
In this paper, a new beam model for the simulation of static bending, free vibration and buckling behaviour of variable thickness FGM beams is proposed. This new beam model modifies the first order shear deformation theory of Timoshenko’s beam by assuming the total deflection as the sum of bending and transverse shear deflections, the angle of cross-section slope as the sum of pure bending and shear angles. Numerical analyses are carried out to investigate the effect of small parameter, degree of non-uniformity, power-law material index and length-to-height ratio on the mechanical behaviour of the variable thickness FGM beams. Simply supported, clamped and free boundary conditions are considered. The numerical results are validated by means of comparisons with data retrieved from the pertinent literature.
The paper is well organized, the mathematical formulations of the equations of motion are correctly reported, the numerical results are properly validated by means of comparisons, several interesting parametric studies are performed by varying mechanical and geometric characteristics.
In order to consider this paper for publication, there are two relevant aspects that should be carefully taken into account by the Authors.
1) Several major English Language mistakes are present throughout the text, which must be absolutely corrected in the reviewed version of the paper. For example, the title of the paper is not clear, and it could be modified in the following way: “A new beam model for the simulation of the mechanical behaviour of variable thickness functionally graded material beams based on a modified first order shear deformation theory”.
2) The Introduction does not provide sufficient background and therefore it must be improved. In fact, the Introduction, in the form as it is written, reports only a list of references related to the topic presented in the paper, without describing open questions or industrial applications of the FGM beams. As an example, in the opinion of the Reviewer, it should be emphasized the importance of the multi-layered functionally graded beams in reducing the mechanical vibrations of composite industrial components by increasing the damping behaviour at the layer interfaces. Therefore, the following relevant papers should be added in the Introduction and discussed:
a) Catania, G.; Strozzi, M. Damping oriented design of thin-walled mechanical components by means of multi-layer coating technology. Coatings 2018, 8, 73.
b) Demir, E. Vibration and damping behaviors of symmetric layered functional graded sandwich beams. Structural Engineering & Mechanics 2017, 62(6), 771-780.
Many other industrial applications of the present work could be added by the Authors.
Therefore, in the opinion of the Reviewer, by considering the previous notes, the paper should be accepted for publication after major revision.
Author Response
Thank you for your comments, Please find in the attachment our point-by-point response

Reviewer 2 Report
This manuscript represents the development of a modified shear deformation beam theory applied to model beam elements with variable thickness based on FEM. The structure of the paper is well organized and the overall paper tells a logical story with a concrete conclusion. However, there are some issues that the author must be addressed to improve the paper:
The motivation is not clear and what's your scientific contribution?
Introduction section must be improved with previous works related to the research topic
A comparative with current numerical formulations related to shear and volumetric locking (u/p formulation, incompatible modes, hourglassing,...) should be included in order to highlight your contributions
More complex numerical simulations should be included using commercial FEM software to show the performance of the proposed model
Author Response

(The authors gave the same response as above.)

Round 2
Reviewer 1 Report
The Authors correctly made all the changes indicated in the previous review. Therefore, in the opinion of the Reviewer, this revised version of the paper is acceptable for publication in the Journal Materials.
Reviewer 2 Report
Accept in present form